# Microencapsulation of Copper(II) Sulfate in Ionically Cross-Linked Chitosan by Spray Drying for the Development of Irreversible Moisture Indicators in Paper Packaging

**DOI:** 10.3390/polym12092039

**Published:** 2020-09-08

**Authors:** Sandra Rojas-Lema, Jorge Terol, Eduardo Fages, Rafael Balart, Luis Quiles-Carrillo, Cristina Prieto, Sergio Torres-Giner

**Affiliations:** 1Technological Institute of Materials (ITM), Universitat Politècnica de València (UPV), Plaza Ferrándiz y Carbonell 1, 03801 Alcoy, Spain; sanrole@epsa.upv.es (S.R.-L.); luiquic1@epsa.upv.es (L.Q.-C.); 2Textile Industry Research Association (AITEX), Plaza Emilio Sala 1, 03801 Alcoy, Spain; Jorgetd92@gmail.com (J.T.); efages@aitex.es (E.F.); 3Novel Materials and Nanotechnology Group, Institute of Agrochemistry and Food Technology (IATA), Spanish National Research Council (CSIC), Calle Catedrático Agustín Escardino Benlloch 7, 46980 Paterna, Spain; cprieto@iata.csic.es

**Keywords:** copper, chitosan, cellulose, ionic cross-linking, spray drying, humidity sensors, intelligent packaging

## Abstract

Copper(II) sulfate-loaded chitosan microparticles were herein prepared using ionic cross-linking with sodium tripolyphosphate (STPP) followed by spray drying. The microencapsulation process was optimal using an inlet temperature of 180 °C, a liquid flow-rate of 290 mL/h, an aspiration rate of 90%, and an atomizing gas flow-rate of 667 nL/h. Chitosan particles containing copper(II) sulfate of approximately 4 µm with a shrunken-type morphology were efficiently attained and, thereafter, fixated on a paper substrate either via cross-linking with STPP or using a chitosan hydrogel. The latter method led to the most promising system since it was performed at milder conditions and the original paper quality was preserved. The developed cellulose substrates were reduced and then exposed to different humidity conditions and characterized using colorimetric measurements in order to ascertain their potential as irreversible indicators for moisture detection. The results showed that the papers coated with the copper(II) sulfate-containing chitosan microparticles were successfully able to detect ambient moisture shown by the color changes of the coatings from dark brown to blue, which can be easily seen with the naked eye. Furthermore, the chitosan microparticles yielded no cytotoxicity in an in vitro cell culture experiment. Therefore, the cellulose substrates herein developed hold great promise in paper packaging as on-package colorimetric indicators for monitoring moisture in real time.

## 1. Introduction

Intelligent packaging has been categorized both as a part of active packaging and, more recently, as a separate entity [1]. It can be defined as packaging types that can sense environmental changes and monitor the condition of packaged goods and/or the environment surrounding the goods. Therefore, it basically provides information about the state of the product to enhance convenience for manufacturing and distribution as well as to improve security and safety verification. Depending on the information provided, intelligent packaging is habitually classified into two main groups, namely, systems that act as data storage (e.g., smart labels) and those that perform as external indicators of different aspects of the product [2]. The latter group is quite broad and includes time–temperature indicators (TTIs) as well as indicators for freshness, ripeness, thawing, moisture, leakage, etc. For instance, tracking the condition of a package during its transportation through the whole supply chain can guarantee that the goods have not been exposed to the wrong conditions. In food technology, self-reading indicators can also help the consumer better judge food quality to avoid disposal of still-fresh food based merely on an estimated date [3].

In paper packaging applications, the use of sensors that perform as indicators in monitoring moisture becomes particularly relevant since cellulose is a highly hygroscopic material and its strength is considerably reduced when exposed to high relative humidity (RH) levels [4]. The moisture content, that is, the amount of water in paper, is not only determined by the RH level present but it also rather depends on the history of the RH changes in the packaging surroundings [5]. In this regard, there is currently a motivation in scientists to develop colorimetric detection methods because of their simplicity, rapidity, and especially their lack of need for expensive instruments [6]. At present, several efforts have been made to improve the sensing performance of cellulose paper-based materials by introducing organic molecules, loading metal particles or grafting polymer structures [7]. In recent times, there has been an increase in the use of copper as a material sensitive to humidity changes since it presents different colors depending on the oxidation state [8]. For instance, copper(II) sulfate, also known as copper sulphate, comprises a family of inorganic compounds with the chemical formula CuSO_4_(H_2_O)_x_, where *x* ranges from 0 to 5. Pentahydrate (x = 5) is the most commonly encountered salt, which shows a bright blue color and is termed as “blue vitriol” [9]. When copper(II) sulfate pentahydrate is heated at 100–150 °C, it is turned into its anhydrous form, which is a white solid, and develops high hygroscopic properties. Therefore, copper(II) sulfate shows a great deal of potential to be used as a colorimetric sensor for indicating the moisture conditions present inside industrial paper-based packaging, pointing out any penetration of moisture into the packaging and revealing the ambient conditions present during transportation.

Colorimetric sensors can be used in the form of vesicles, thin films, and membranes attached to the indicator support [10,11]. Encapsulation of copper(II) sulfate in a biopolymer matrix can represent a feasible and sustainable alternative, which allows the crystal size to be reduced and thus increases the contact area with the environment [12]. Furthermore, it can serve as the adhesion material to the indicator substrate. Among the potential encapsulating matrices, chitosan has become one of the most widely used biopolymers in recent times. This natural copolymer is composed of β-(1,4)-glucosamine and β-(1,4)-acetylglucosamine, obtained from the deacetylation of chitin, and it is non-toxic, biocompatible, biodegradable, bioadhesive, and a natural polycationic agent [13]. In particular, chitosan has been described as a suitable biopolymer for the collection of metal ions since its amino and hydroxyl groups can act as chelation sites for metal ions [14]. For instance, adsorption of Cu^2+^ ion concentrations below 1 mg/L can be achieved after long exposure times (up to 3 days) [15,16]. Furthermore, the chelation of copper ions with chitosan has received considerable attention [17] and results indicated that the activity of chitosan–metal complexes can depend on the property of metal ions, the molecular weight (M_W_), and the degree of deacetylation (DD) of the biopolymer and environmental pH values [18].

Since chitosan is hydrophilic and easily swells in aqueous media, cross-linking is habitually required to avoid undesired burst release, improve the mechanical properties, and obtain better shape microparticles while avoiding agglomeration [19]. However, typical chitosan cross-linking with glutaraldehyde may induce undesirable toxic effects. For example, glutaraldehyde can cause irritation to mucosal membranes because of its toxicity [20]. To overcome the disadvantage of chemical cross-linkers, other cross-linkers (e.g., genipin) have been proposed, which offer 5000‒10,000 times less cytotoxicity than glutaraldehyde [21,22], while ionic cross-linking interaction has also been explored. In the latter case, sodium tripolyphosphate (STPP) is a nontoxic polyanion that interacts with chitosan in acidic medium via electro-static forces to form ionically cross-linked networks [23]. Moreover, the unique polycationic structure of ionic cross-linkers allows an optimal use of chitosan for drug delivery [24]. Indeed, STPP is classified as generally recognized as safe (GRAS) by the Food and Drug Administration (FDA) [25].

Chitosan microparticles can be prepared using different procedures depending on the kind of product to be encapsulated and the targeted application. Among them, ionic gelation, spray drying, electrospraying, emulsification-solvent evaporation, and coacervation have showed notable relevance [26,27]. STPP-cross-linked chitosan micro- or nanoparticles have been produced using either the emulsion or syringe method [28,29,30]. However, since these methods involve tedious processes and do not yield reproducible results, they could be unsuitable for large-scale production. In contrast, spray drying is nowadays widely used in the pharmaceutical and food industry for the preparation of microparticles since it is a one-stage continuous process, easy to scale up, and with a relatively low processing cost [31]. This process consists of spraying a solution inside a chamber at high temperature [32]. When the liquid is fed into the nozzle through a peristaltic pump, atomization occurs due to the force of the compressed air disrupting the liquid into small droplets. The solvent in the droplets is evaporated by hot air, and the dried particles are separated using a cyclone.

The aim of this work was to prepare microcapsules of chitosan containing copper(II) sulfate and carry out their preliminary assessment for the development of a colorimetric moisture indicator with application interest in intelligent paper packaging. To this end, copper(II) sulfate was first encapsulated in chitosan microparticles using ionic gelation with STPP followed by spray drying. The effect of the spray drying parameters on the production yield, moisture content of the microcapsules, and their morphology was analyzed. After that, the most optimal microcapsules were deposited on a cellulose substrate and fixed via two different methods, namely cross-linking by means of STPP or using a chitosan hydrogel. Finally, the potential of the indicator was evaluated by subjecting the sensor to different moisture conditions and quantifying the color change. In addition, cytotoxicity tests were performed on the microcapsules to confirm the non-toxicity of the resultant sensor for industrial applications.

## 2. Materials and Methods

### 2.1. Materials

Chitosan, copper(II) sulfate, STPP, 2-hydroxypropyl-β-cyclodextrin, dimethyl sulfoxide (DMSO), and sodium borohydride (NaBH_4_) were all purchased from Sigma Aldrich S.A. (Madrid, Spain). Acetic acid was obtained from VWR Chemicals (Llinars del Vallés, Spain). Dubelcco’s modified Eagle’s medium (DMEM) and Invitrogen^TM^ Glutamax was purchased at Thermo Fisher Scientific (Waltham, MA, USA). Fetal bovine serum (SBF) of 10 wt % was delivered by Biochrom Ltd. (Holliston, Massachusetts, MA, USA). A colorimetric kit of cellular proliferation 3-(4,5-dimethylthiazole-2-yl)-2,5-diphenyltetrazolium bromide (MTT) assay was obtained from Hoffmann-La Roche (Basel, Switzerland). A cellulose non-woven mat with a surface density of 300 g/m^2^ provided by AITEX (Alcoy, Spain) was used as the paper substrate.

### 2.2. Preparation of the Copper(II) Sulfate-Loaded Chitosan Particles

Chitosan was first dissolved in acetic acid 1% (vol/vol) at a concentration of 0.5 wt %. The solution was left under stirring for 24 h at room temperature to achieve total dissolution. After that, copper(II) sulfate was added at a rate of 2:1 (wt/wt) in relation to chitosan. The resultant solution was left under stirring for 10 min at room temperature. A STPP solution was prepared in deionized water at a concentration of 10 wt %. Chitosan colloids were spontaneously fabricated after dropwise addition of the aforementioned STPP solution to the chitosan solution at a chitosan-to-STPP ratio of 2:1 (wt/wt), under magnetic stirring for 30 min at room temperature. This chitosan-to-STPP ratio was selected based on previous studies based on saturation of amino groups in chitosan using STPP in acidic conditions, since it provided an opalescent solution, which is representative for the optimum cross-linking conditions [33,34].

The chitosan particles were hardened and dried using a Mini Spray Dryer B-290 (Buchi Laboratoriums-Tecnik, Flawil, Switzerland). The adjustment of the operation parameters that ensured a stable and robust process was made experimentally. The operation parameters varied as follows: inlet air temperature in the range 160–220 °C, liquid flow-rate between 213 and 363 mL/h (15–25%), atomizing gas flow-rate between 439 and 667 nL/h (3–4 cm in the rotameter), and aspiration rate in the range 70–90%. The diameter of the nozzle used was 0.7 mm. The equipment operated in co-current flow mode. In order to maintain homogeneity, the colloidal solution was kept under magnetic stirring during the spray drying process. The solid capsules were harvested from the apparatus collector and stored under vacuum at room temperature. Production yield was expressed as the ratio between the weight percentage of the final product versus the total amount of the materials sprayed. Figure 1 shows the spray-dryer unit with the operation parameters.

### 2.3. Microscopy

The morphology of chitosan capsules was analyzed using focused ion beam scanning electron microscopy (FIB-SEM) in a Zeiss Neon 40 (Jena, Germany) with an electron beam acceleration of 2 kV. The samples were coated with a gold/palladium layer prior to FIB-SEM analysis. Particle diameters were determined using Image J Launcher v1.41 (National Institutes of Health, Bethesda, MD, USA), and the data presented were based on measurements from a minimum of 20 micrographs.

### 2.4. Moisture Determination

The moisture content of the chitosan capsules was determined using gravimetric measurement performed in triplicate. To this end, 0.5 g of chitosan capsules were placed in a vacuum oven (JP Selecta, Barcelona, Spain) at 70 °C and −0.1 bar and their mass was recorded until a constant weight was reached. The moisture content was determined from the difference between the initial mass of the sample (*m_i_*) and the final one (*m_f_*) using Equation (1):(1)$$Moisture\;\left(\%\right)=\frac{m_{i}-m_{f}}{m_{f}}*100$$

### 2.5. Cytotoxicity Tests

Cytotoxicity of the chitosan microcapsules was evaluated due to the presence of STPP in the formulations and also the use of copper(II) sulfate. The extractive method was used to analyze the samples according to the ISO 10993-5:2009 standard [35]. Different contents of copper(II) sulfate-loaded chitosan microcapsules in STTP were assayed as described in Table 1. The ratios of STPP and chitosan microcapsules ranged between 5 and 20 mL of STPP per gram of chitosan using two different concentrations of chitosan in solution, that is, 0.5 and 1 wt %. L929 mouse fibroblasts were used as cell lines and cultured in 96-well plates using DMEM, glutamax, and SBF as culture medium. The extract of the cross-linked materials was placed over the cells. The contact time between the samples and cell lines was 24 h, and cell viability was quantified using MTT assay and spectrophotometry. The mitochondrial activity, which is directly proportional to the number of viable cells, was measured in a Multiskan Ascent microplate reader (MTX Lab Systems, Bradenton, FL, USA). A solution of DMSO at 5 wt % in DMEM and SBF was used as positive cytotoxic control. The analysis was performed in triplicate.

### 2.6. Fixation of the Copper(II) Sulfate-Loaded Chitosan Particles

Figure 2 shows the methodology followed to incorporate the chitosan microcapsules containing copper(II) sulfate into the paper substrate in the form of coatings. This process was carried out using two methods: ionic cross-linking (Figure 2a) and the formation of a chitosan hydrogel (Figure 2b). For the first method, the chitosan particles were uniformly spread over the substrate, a solution of STPP at 25 wt % was sprayed, and the coated substrate was left in the oven at 60 °C for 1 min. In the second method, 100 g of a chitosan solution at 1 wt % in acetic acid was stirred for 24 h at room temperature and the resultant solution was manually spread over the substrate. When the chitosan hydrogel started drying, the capsules were spread and left until drying was complete. Different drying times were assessed.

### 2.7. Colorimetric Measurements

A solution of NaBH_4_ in water at 10 wt % was spread directly over the substrate containing the chitosan capsules until they reached a dark brown color. Thereafter, the substrates were placed at four different humidity conditions, that is, 0%, 15%, 60%, and 100% RH, which resemble dissimilar packaging conditions, for 24 h to study the color change. The colorimetric determination of the substrates was carried out in a benchtop spectrophotometer HunterLab ColorFlex EZ colorimeter from Hunter Associates Laboratory Inc. (Reston, VA, USA). The Commission Internationale de L’Eclairage (CIE) standard illuminant D65 was used to assess the CIE Lab color space coordinates L*a*b* using an observer angle of 10° in which L* represents the luminance (black to white), a* indicates the change between green and red, and b* represents the change from blue to yellow. The colorimeter was calibrated with a white standard tile and a mirror device for black (no light reflection). An average of 20 measurements per sample were taken.

## 3. Results and Discussion

### 3.1. Optimization of the Spray Drying Process

Ionic cross-linking of chitosan was performed in order to develop water-resistant capsules after spray drying. This process consisted of the spontaneous reaction of cationic chitosan with the anionic STPP cross-linking agent according to the reaction scheme shown in Figure 3. During this process, chitosan is first solubilized in acetic acid and the addition of the STPP solution results in a polyelectrolyte complex, stabilized by a cross-linked electrostatic interaction between the NH_3_^+^ groups of chitosan and the STPP-O^−^ groups. After this, a three-dimensional entanglement is precipitated from an aqueous solution in the form of gel-like particles. The reaction was performed in diluted acid solution since the glucosamine groups are pH-sensitive (pKa ≈ 6), where the glucosamine units are converted into a soluble form of protonated amine (R–NH_3_^+^) [36].

Spray drying was used to convert the copper(II) sulfate-containing chitosan-STPP colloidal solution into a dry powder with a relatively narrow particle size distribution. The physical properties of the resulting product, that is, the particle morphology and moisture content, were adjusted through the manipulation of the spray drying process variables. The main operating parameters include the inlet temperature, liquid flow-rate, atomizing gas flow-rate, and aspiration rate. As shown in Table 2, these parameters were varied in order to both maximize production yield and minimize moisture content in the resultant powder. Production yield may compromise the economic viability of the process, whereas moisture content of the powder could affect the stability of the indicator and also the morphology of the capsules.

It can be observed that, at all the experimental conditions, spray drying showed a relatively low production yield of chitosan capsules due to the powder adhering to the cyclone walls. However, it was observed that the use of lower inlet temperatures, between 160 and 180 °C, significantly favored the production yield, with an increase of up to 51%. Moreover, higher air atomizing flow-rates produced an increase in the production yield of approximately 15%. Furthermore, increasing the liquid flow-rate led to a decrease in the production yield. Finally, the higher the aspiration rate, the higher the production yield, being slightly higher for an aspiration rate of 80%.

Regarding the moisture content of the chitosan capsules, one can observe that higher temperatures favored water evaporation, and consequently, lower values were attained. A similar observation can be made by reducing the liquid flow-rate. By increasing the flow-rate, the drying capacity of the equipment is surpassed, obtaining higher moisture content in the particles. At higher aspiration rates, samples showed a higher moisture content as a consequence of the reduction in residence time inside the drying chamber. Finally, the moisture content decreased by increasing the atomizing gas flow-rate. It can be expected that by increasing the atomizing gas flow-rate, smaller droplets with a larger surface area are generated and, thus, they evaporate faster.

Therefore, the operating conditions that maximize production yield and a reduced moisture content were an inlet temperature of 180 °C, liquid flow-rate of 290 mL/h, aspiration rate of 90%, and atomizing gas flow-rate of 667 nL/h. These parameters are in agreement with those previously reported by other authors to encapsulate different types of compounds in chitosan using spray drying, including pharmaceutical compounds, enzymes or antioxidants, among others, where the inlet temperature varied between 120 and 180 °C and the liquid flow-rate between 120 and 420 mL/h [37]. Additionally, Helbling et al. [38] reported a maximum operation yield when encapsulating progesterone into STPP-cross-linked chitosan using spray drying, using an inlet temperature of 170 °C and a liquid flow-rate of 204 mL/min. Similarly, Learoyd et al. [39] selected an inlet temperature of 180 °C, a liquid flow-rate of 192 mL/h, an atomizing gas flow-rate of 600 L/h, and an aspiration rate of 85% for the encapsulation of beclometasone dipropionate in chitosan using spray drying, obtaining encapsulation efficiencies over 60%.

### 3.2. Morphological Characterization of the Chitosan Microcapsules

Figure 4 shows a SEM micrograph of the copper(II) sulfate-loaded chitosan particles obtained via spray drying at the aforementioned optimal operating conditions. Shrunken microparticles with a medium particle size of approximately 4 µm and a wide particle size distribution were obtained. The roughness of the surface of the microparticles can be related to the rapid evaporation of the solvent and the formation of an external crust during the first stages of drying, which collapses when the solvent present in the inner parts of the droplet evaporates and leads to a partial shrinkage of the particle [26]. In addition, the use of STPP for chitosan cross-linking could also contribute to an increase in the surface roughness [37]. Similarly, Helbling et al. [38] observed that the addition of STPP favored the formation of shrunken microparticles with a mean size of 4 µm and a raisin morphology, when encapsulating progesterone into STPP-cross-linked chitosan using spray drying with an inlet temperature of 170 °C and a liquid flow-rate of 204 mL/min. In another study, Aranaz et al. [31] also obtained shrunken microparticles by encapsulating venlafaxine hydrochloride into STPP-cross-linked chitosan using spray drying with an inlet temperature of 160 °C. Furthermore, as demonstrated by Kašpar et al. [32], the raisin morphology could not be avoided by performing the cross-linking step combined with the spray drying process using a coaxial nozzle.

Figure 5 shows a comparative scheme of the particles’ morphologies obtained under different operating conditions. It was observed that increasing the liquid flow-rate exerted two effects on the morphology of the microparticles. On the one hand, it increased the particle size and yielded a wider particle size distribution. On the other, the number of indentations on the microparticle surface was reduced due to the slower evaporation rate. Regarding the atomizing gas flow-rate, increasing this parameter led to smaller chitosan particles since smaller droplets were generated in the nozzle. One can also notice that the increase in the aspiration rate did not show any significant morphological effect on the chitosan particles in the studied range.

However, as shown in Figure 6, the increase in inlet temperature favored the formation of shrunken-type microparticles due to a faster evaporation rate. Similar observations were previously made by other authors [38,40]. For instance, Wei et al. [40] recently observed that the particle size increased when increasing the liquid flow-rate, whereas the number of indentations increased by increasing the inlet temperature, during the encapsulation of theopylline in STPP-cross-linked chitosan using spray drying.

### 3.3. Cytotoxicity of the Chitosan Microcapsules

Cytotoxicity of the copper(II) sulfate-containing chitosan microcapsules was evaluated in order to ascertain its potential in packaging and textile applications. Figure 7 shows the cell viability of the different solutions of copper(II) sulfate-loaded chitosan microparticles in STPP. The cytotoxicity tests carried out with the samples containing 0.5 wt % of chitosan microparticles, that is, samples from 1 to 4, indicated that none of the tested solutions showed cytotoxicity (see Figure 7a). In these cases, cell viability was in the range of the control sample, that is, a solution medium without chitosan and STPP. In the case of the samples prepared with higher content of chitosan microcapsules, that is, 1 wt %, which were labelled as samples 5 to 8 and shown in Figure 7b, the cell viability of the samples slightly decreased at the highest STTP/chitosan ratios, that is, 15 and 20 mL/g. Nevertheless, in all cases, the cell viability was greater than 70%. Similarly, in vitro culture of human fibroblasts performed by Liu and Gao [41] to assess the biocompatibility of chitosan nanoparticles ionically cross-linked by STPP showed that no difference in the cytoviability was found between the cells cultured in the medium containing chitosan nanoparticles and the fibroblasts cultured in the control medium at all the culture times. In the study of Gritsch et al. [42], dealing with the antibacterial properties of copper(II)-chitosan complexes for potential uses in biomedicine, it was also indicated that it is possible to avoid cytotoxicity by fine tuning the amount of copper.

### 3.4. Fixation of the Chitosan Microcapsules to the Paper Substrate

Figure 8 presents the optical images of the copper(II) sulfate-containing chitosan microcapsules deposited on the cellulose substrates and fixated via cross-linking with STPP and by the use of a chitosan hydrogel. It was observed that the substrate fixated by means of the cross-linking method, shown in Figure 8a, led to weak adhesion and the microcapsules easily detached. Furthermore, it can also be observed that the use of STPP visually contaminated the cellulose substrate during the fixation process and it lost its original physical–chemical properties, such as uniformity and smoothness. Alternatively, as it can be observed in Figure 8b, the addition of the chitosan microcapsules using a hydrogel support successfully yielded a coating strongly adhered to the paper substrate. Furthermore, the visual appearance of the paper substrate remained unaltered. Nevertheless, drying time played a main role in assuring optimal adhesion and also avoiding the damage of the porous structure of paper. In particular, it was observed that the best fixation was achieved for a drying time of approximately 8 min when 100 g of chitosan microcapsules were deposited. Since the main force between cellulose fibers is a hydrogen bond, one can consider that the mild conditions applied herein were appropriate for preserving the original paper quality.

### 3.5. Evaluation of the Paper Substrates as Moisture Indicators

The colorimetric characterization of the samples was performed on the paper substrates in which the copper(II) sulfate-loaded chitosan microcapsules were fixed via the chitosan hydrogel support since it provided the best performance in terms of adhesion and paper quality. The substrates containing the chitosan microcapsules were thereafter reduced by wise dropping the aqueous NaBH_4_ solution to yield copper metal, which is a reddish brown solid. As described by Glavee et al. [43], metallic copper plus hydrogen gas is formed in the redox reaction between copper ions with NaBH_4_ following the reaction scheme shown in Equation (2):2 Cu^+2^_(aq)_ + 4 BH^−^_4 (aq)_ + 12 H_2_O → 2 Cu^0^_(s)_ + 14 H_2 (g)_ + 4 B(OH)_5_(2)

The process above led first to a yellow-to-brown material, followed by immediate gas evolution with the final formation of a dark brown solid that is characteristic of metallic copper, which was formed instead of the boride due to the positive redox potential of copper.

Table 3 presents the color parameters of the substrates as a function of the %RH exposed for 24 h. Values of RH of 0%, 15%, 60%, and 100% were selected as being representative for dry, intermediate, and wet packaging conditions. One can observe that, at dry conditions, that is, 0% RH, the substrate presented color parameters typical of copper metal, having a* and b* values of 3.08 (red) and 9.55 (yellow), respectively. When the paper substrates were exposed to moisture, L increased slightly, while the a* and b* color coordinates shifted to lower and higher values, respectively. For instance, when the paper substrate was subjected to 60% RH, it showed a value of L of 60.01 (darker) and a* and b values of −18.76 (green) and 13.39 (yellow). Therefore, the blue hue of the paper substrates increased progressively with the %RH in the environment, confirming that the developed coating based on copper was highly sensitive to moisture at damp conditions.

Figure 9 shows the visual aspect of the paper substrates after reduction and prior to moisture exposure (Figure 9a) and after being exposed at RH of 60% for 24 h (Figure 9b), confirming their potential as colorimetric indicators for moisture detection since color changes can be easily seen with the naked eye. The color change of the paper substrates, before and after exposure to moisture, is related to the oxidation reaction between copper metal, oxygen, and water, which can be described using Equations (3) and (4) [8]:2 Cu^0^_(s)_ + O_2 (g)_ + 2 H_2_O →2 Cu^+2^_(aq)_ + 4 OH^−^_(aq)_(3)
2 Cu^0^_(s)_ + 2 H_2_O → 2 Cu^+2^_(aq)_ + H_2 (g)_ + 2 OH^−^_(aq)_(4)

In the hydrated substrates, water molecules act as ligands providing free electrons to Cu^2+^ ion (from oxygen atom of water), and the blue color is due to the d-d transition for the d9 configuration of Cu(II) [44]. Since the reduction process to attain copper metal has to be performed prior to incorporating the substrate into the packaging system, the resultant indicator becomes irreversible once the sensor is oxidized and the color change is produced in real time.

The amount of published work on moisture indicators for packaging applications is still limited. Some trials, however, have been performed to develop colorimetric humidity-indicating sensors based on copper. For instance, indicating agents of interest in humidity-adsorbing columns were recently prepared by Ulutan et al. [45] by drying silica hydrogels impregnated with a saturated copper sulfate solution at 100 °C. The prepared dried copper sulfate-containing gels containing light blue CuSO_4_·H_2_O were transformed upon moisture adsorption at 25 °C into dark blue CuSO_4_·3H_2_O, as clearly shown using visible spectroscopy.

## 4. Conclusions

Cellulose has the various advantages of low cost, good portability, excellent flexibility, and biocompatibility for several applications including paper packaging and textiles. However, paper quality is dependent on moisture conditions since cellulose fibers are highly hydrophilic. The present study aimed to microencapsulate copper(II) sulfate in ionically cross-linked chitosan using STPP and perform a preliminary assessment of the resultant microcapsules as on-package sensors for real-time moisture detection in the packaging environment. Spray drying was selected as the encapsulation process since it is a reproducible, rapid, and relatively easy to scale up. The encapsulation process was optimized in order to make it as efficient as possible and generated the capsules with the lowest moisture contents. The best conditions were attained for an inlet temperature of 180 °C, a liquid flow-rate of 290 mL/h, an aspiration rate of 90%, and an atomizing gas flow-rate of 667 nL/h. These conditions successfully yielded copper(II) sulfate-containing capsules of chitosan of approximately 4 µm with a shrunken-type morphology. Furthermore, it was observed that the copper(II) sulfate-loaded microcapsules, at different STPP-to-chitosan ratios, were not cytotoxic. Among the fixation methods, the formation of a chitosan hydrogel led to the most optimal support for the incorporation of the copper(II) sulfate-containing chitosan microcapsules into the paper substrate since this was performed at milder conditions and the original paper quality was preserved. Finally, the resultant paper substrates coated with the copper(II) sulfate-containing microcapsules were evaluated at different moisture conditions, from 0% to 100% RH, yielding a color change from dark brown to blue. From the above, it can be considered that the here-developed paper substrates can find wide applicability in the packaging and textile fields as irreversible humidity indicators.

## Figures and Tables

**Figure 1 polymers-12-02039-f001:**
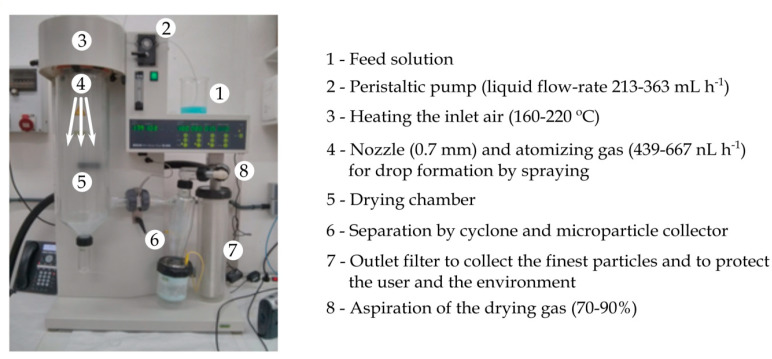
Setup of the spray-dryer unit with indications of its components and working procedure.

**Figure 2 polymers-12-02039-f002:**
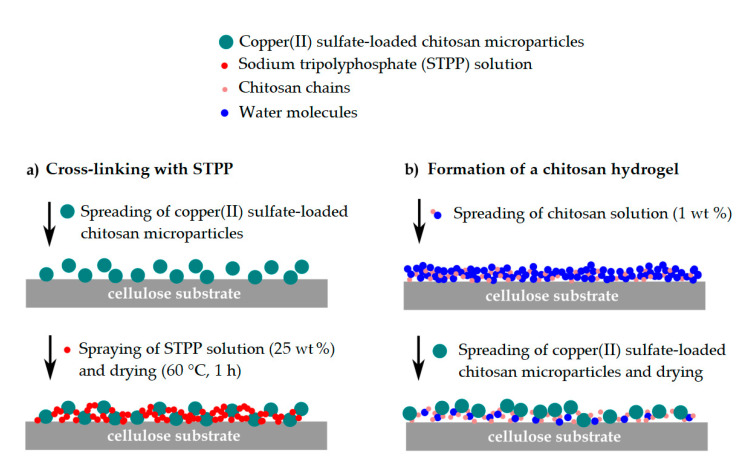
Schematic procedure for the fixation of the copper(II) sulfate-loaded chitosan microparticles on the cellulose substrate by (**a**) ionic cross-linking with sodium tripolyphosphate (STPP) and (**b**) formation of a chitosan hydrogel.

**Figure 3 polymers-12-02039-f003:**
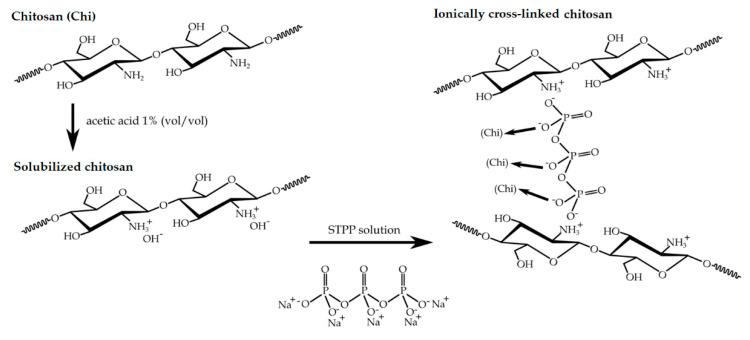
Ionic cross-linking reaction of chitosan using sodium tripolyphosphate (STPP).

**Figure 4 polymers-12-02039-f004:**
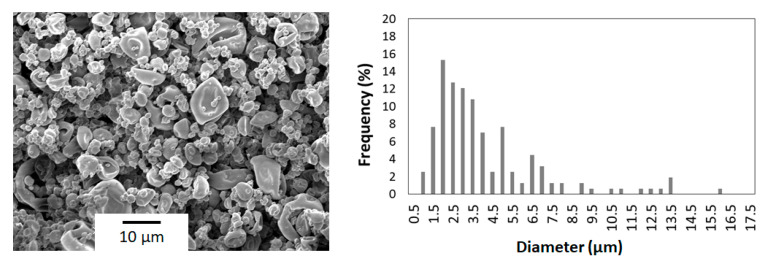
Scanning electron microscopy (SEM) micrograph and diameter histogram of the copper(II) sulfate-loaded chitosan microparticles obtained for an inlet temperature of 180 °C, liquid flow-rate of 290 mL/h, aspiration rate of 90%, and atomizing gas flow-rate of 667 nL/h.

**Figure 5 polymers-12-02039-f005:**
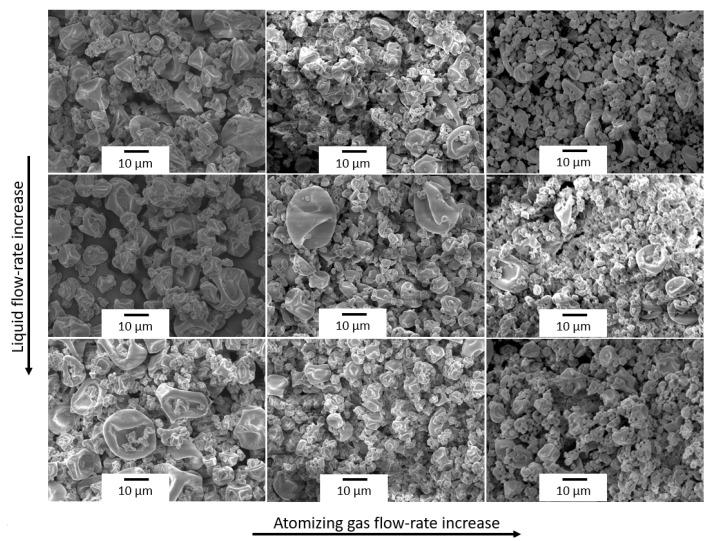
Comparative scheme of the scanning electron microscopy (SEM) micrographs of the copper(II) sulfate-loaded chitosan microparticles obtained at 180 °C and a constant aspiration rate of 90%. The columns represent the atomizing gas flow-rates (439, 538, and 667 nL/h) and the rows represent liquid flow-rates (213, 290, and 363 mL/h). Scale markers of 10 µm.

**Figure 6 polymers-12-02039-f006:**
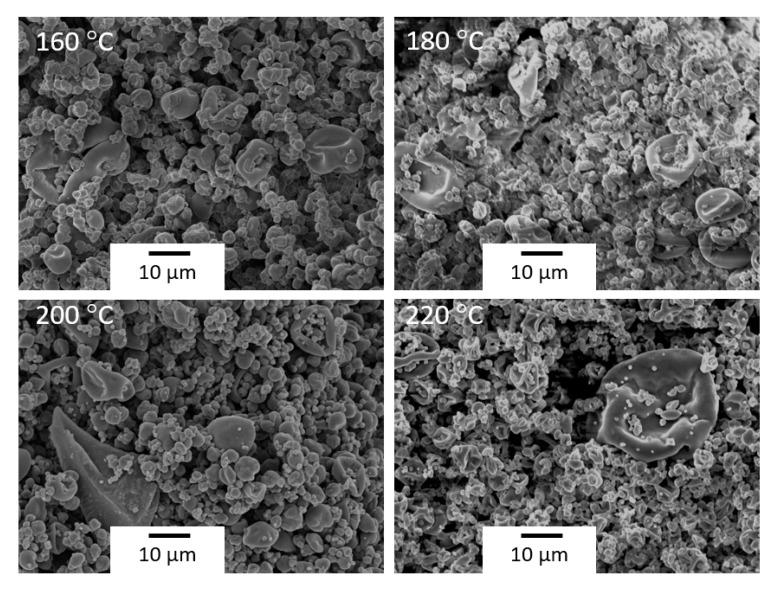
Comparative scheme of the scanning electron microscopy (SEM) micrographs of the copper(II) sulfate-loaded chitosan microparticles obtained at different inlet temperatures, for a constant aspiration rate of 90%, atomizing gas flow-rate of 667 nL/h, and liquid flow-rate of 290 mL/h. Scale markers of 10 µm.

**Figure 7 polymers-12-02039-f007:**
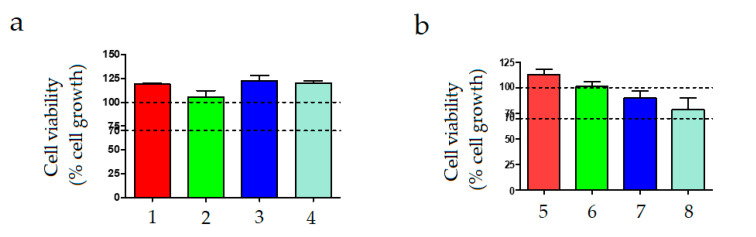
Cell viability in terms of percentage (%) of cell growth with respect to the control of L929 cell lines in contact with solutions of copper(II) sulfate-loaded chitosan microcapsules in sodium tripolyphosphate (STPP) for 24 h of exposure at chitosan contents of: (**a**) 0.5 wt %; (**b**) 1 wt %.

**Figure 8 polymers-12-02039-f008:**
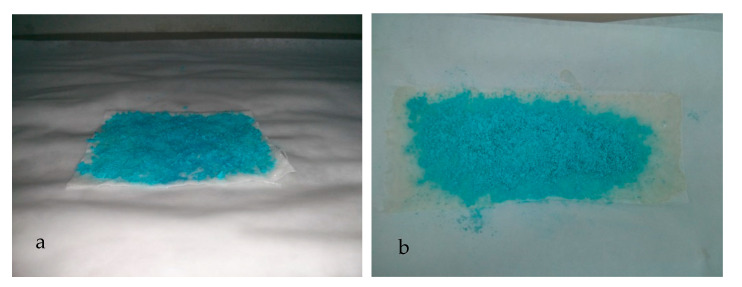
Paper substrates with the different copper(II) sulfate-containing chitosan microcapsules fixated via (**a**) cross-linking with sodium tripolyphosphate (STPP) and (**b**) chitosan hydrogel support.

**Figure 9 polymers-12-02039-f009:**
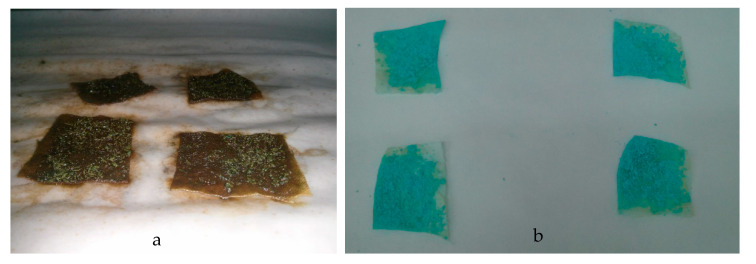
Paper substrates with the different copper(II) sulfate-containing chitosan microcapsules: (**a**) reduced with sodium borohydride (NaBH_4_); (**b**) oxidized at relative humidity (RH) of 60%.

**Table 1 polymers-12-02039-t001:** Set of liquid solutions prepared for the cytotoxicity tests according to the content of copper(II) sulfate-loaded chitosan microcapsules in sodium tripolyphosphate (STPP). All solutions were prepared with a total mass of 50 g.

Sample	Chitosan (g)	STTP (mL)	STTP/Chitosan Ratio (mL/g)
1	0.25	1.25	5
2	0.25	2.50	10
3	0.25	3.75	15
4	0.25	5.00	20
5	0.50	2.50	5
6	0.50	5.00	10
7	0.50	7.50	15
8	0.50	10.00	20

**Table 2 polymers-12-02039-t002:** Yield and moisture content for the different operating conditions evaluated.

Inlet Temperature (°C)	Atomizing Gas Flow-Rate (nL/h)	Liquid Flow-Rate (mL/h)	Aspiration Rate (%)	Yield (%)	Moisture (%)
160	538	290	80	42.73 ± 2.23	6.87 ± 0.32
		363		40.45 ± 3.67	7.99 ± 0.67
		290	90	30.30 ± 3.05	8.43 ± 1.03
		363		43.64 ± 4.14	5.50 ± 0.88
	667	290	80	46.52 ± 3.54	11.34 ± 0.98
		363		45.61 ± 4.70	7.33 ± 0.72
		290	90	46.82 ± 2.46	10.02 ± 0.54
		363		52.57 ± 2.20	6.17 ± 0.33
180	439	213	70	26.97 ± 2.01	11.30 ± 0.27
		290		23.09 ± 2.24	15.24 ± 1.12
		363		40.91 ± 1.87	10.95 ± 0.98
		213	80	40.00 ± 2.26	11.50 ± 1.36
		290		46.17 ± 3.03	10.49 ± 0.74
		363		26.29 ± 1.23	21.24 ± 0.54
		213	90	33.37 ± 1.41	10.40 ± 0.75
		290		36.57 ± 1.20	11.49 ± 1.00
		363		25.37 ± 1.96	13.45 ± 1.03
	538	213	70	52.80 ± 3.08	11.61 ± 0.98
		290		46.63 ± 2.79	9.15 ± 0.54
		363		44.11 ± 3.25	9.79 ± 0.76
		213	80	49.60 ± 2.68	9.29 ± 1.12
		290		48.00 ± 3.10	9.33 ± 0.97
		363		45.71 ± 2.41	8.15 ± 1.08
		213	90	69.26 ± 4.20	7.99 ± 0.44
		290		38.17 ± 2.98	6.81 ± 0.77
		363		39.31 ± 3.02	10.29 ± 0.36
	667	213	70	34.06 ± 1.87	11.09 ± 0.33
		290		49.83 ± 3.12	9.72 ± 0.78
		363		33.00 ± 2.98	12.46 ± 1.54
		213	80	67.20 ± 1.00	8.99 ± 0.89
		290		72.00 ± 3.32	8.89 ± 0.21
		363		62.63 ± 2.49	9.74 ± 1.20
		213	90	70.11 ± 3.88	14.91 ± 0.65
		290		66.97 ± 4.57	7.83 ± 1.01
		363		50.74 ± 3.23	8.67 ± 0.43
200	667	213	80	41.21 ± 2.01	6.03 ± 0.26
		290		39.24 ± 3.22	6.81 ± 0.87
		363		40.45 ± 2.89	7.53 ± 0.22
		213	90	34.39 ± 2.45	7.04 ± 1.04
		290		35.91 ± 1.89	4.86 ± 0.83
		363		34.09 ± 2.40	6.67 ± 1.36
220	667	213	80	39.24 ± 3.02	5.27 ± 0.44
		290		33.64 ± 2.41	7.46 ± 0.80
		363		32.42 ± 1.09	6.54 ± 0.56
		213	90	39.55 ± 2.08	5.83 ± 0.67
		290		44.24 ± 1.67	5.58 ± 0.71
		363		41.67 ± 3.32	5.50 ± 0.58

**Table 3 polymers-12-02039-t003:** Color parameters of the paper substrates stored at different relative humidity (RH, %) for 24 h.

Test	RH (%)	L	a*	b*
1	0	54.33 ± 2.32	3.08 ± 0.49	9.55 ± 0.70
2	15	59.98 ± 1.89	−17.81 ± 1.03	9.63 ± 0.82
3	60	60.01 ± 2.04	−18.76 ± 1.16	13.39 ± 1.75
4	100	61.86 ± 2.04	−25.43 ± 2.07	27.65 ± 1.98

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
