# Peer review of "Microencapsulation of Copper(II) Sulfate in Ionically Cross-Linked Chitosan by Spray Drying for the Development of Irreversible Moisture Indicators in Paper Packaging"

_polymers, 2020, doi:10.3390/polym12092039_

Round 1

Reviewer 1 Report

This manuscript reports a research work on the design and realization of a moisture indicator based on copper sulfate encapsulated in chitosan microparticles. Even if the is well organized, in my opinion, the subject is not new and the experimental results are insufficient for publication. With the exception of few Spray Drying conditions, no one of the main processing parameters (copper sulfate concentration, ambient temperature, and humidity, cellulose substrates conditions/morphology, adhesion properties, etc.) has been investigated. In this sense, additional references both on the theory and experiments related to adhesion properties would improve the level of the work. I also suggest more attention to the cellulose morphology before and after microcapsules deposition. The list of references does not include centrally important and well-known papers in the field. The authors should compare their results with the other studies and report their new insights on this field.

Author Response

According to reviewer’s recommendations, the following issues have been improved in the revised version.

- We have included new relevant references to describe all the scientific phenomena and also compared the results with other studies reporting similar findings or focused on these systems.

- Furthermore, the present study mainly aimed to include results for the microencapsulation of copper(II) sulfate in chitosan and preliminary prove their application in paper packaging. As also added to the text, values of RH of 0%, 15%, 60%, and 100% were selected as being the most representative for dry, intermediate, and wet packaging conditions. Since this was not the original idea of the manuscript and due to the current situation with COVID-19, at this moment we are not able to conduct more experiments. We will, however, prepare a new detailed study regarding the characterization of the substrates (thermal, mechanical, and morphological) in another manuscript as soon as the facilities become available again.

Reviewer 2 Report

The topic is interesting and the results worth of publication. I suggest major revision before publication.

  • The characterization of the material is relatively weak. Additional measurements would increase the impact of this MS. For instance how many cycles of dry/humid atmosphere can be applied? XPS experiments on the oxidation state of Cu could be interesting. What about mechanical performance? This is important for packaging application.
  • The MS should reports results for copper(II) sulfate loaded paper to show the advantage of using chitosan.
  • Figure 1. “size” is it diameter?

Author Response

1) Due to the current situation with COVID-19, we are not able to conduct XPS experiments at this moment. Moreover, although it is correct the mechanical properties are important for packaging applications, due to the use of thin coating, the performance of the paper substrate is not expected to be significantly modified after the particles deposition. Furthermore, one can also consider that the indication process would not be reversible since, as indicated in the Introduction, high temperatures would be need to remove the entrapped water and then turned copper(II) sulfate into its anhydrous form. This was also mentioned in the title and Conclusions.

2) We agree with this comment. The use of chitosan or other matrix would be always necessary to fixate the metal microparticles to the paper substrate. The usefulness of chitosan is based on its high availability and, consequently, its cost and its wide use in the food industry. This has been included in the revised version to assess the usefulness of chitosan polymers.

3) As detected by the reviewer, we have replaced “size” by “diameter”.

Reviewer 3 Report

In this manuscript, authors have demonstarted the microencapsulation of Cu (II) sulfate into ionically crosslinked chitosan by spray drying for the development of moisture indicators in paper packaging applications. This study is interesting and can be considered for publication. However, it needs some improvements as follows:

1) Please incorporate a schematic procedure for the preparation of microparticles and their fixation in the form of figure for better understanding of the method and product.

2) Also, provide the EDX spectra alongwith SEM images.

Author Response

1) As recommended by the reviewer, some schematic figures to depict the procedure for the preparation of the copper(II) sulfate chitosan microparticles, their fixation to the cellulose substrate, and their cross-linking with STPP have been included in the revised version (see Figures 1, 2 and 3, respectively).

2) We are in total agreement with the reviewer since EDX could give interesting information. Nevertheless, due to the current situation with COVID-19, we are not able to conduct more experiments at this moment. In any case, one can expect the metal microparticles to be encapsulated in the chitosan beads since they proved the ability to change color as a function of %RH.

Round 2

Reviewer 1 Report

All my comments were improved in revised version of manuscript. The  manuscript may be considered for publication.

Please, update figure numbers in the new version.

Reviewer 2 Report

In principle the MS is fine. Still the proposed additional experiments would have improved the MS.

Reviewer 3 Report

In my opinion, this manuscript now can be considered for publication.